# Long COVID—six months of prospective follow-up of changes in symptom profiles of non-hospitalised children and young people after SARS-CoV-2 testing: A national matched cohort study (The CLoCk) study

Terence Stephenson[1], Snehal M. Pinto Pereira[2]*, Manjula D. Nugawela[1], Kelsey McOwat[3], Ruth Simmons[3], Trudie Chalder[4], Tamsin Ford[5], Isobel Heyman[1], Olivia V. Swann[6], Lana Fox-Smith[1], Natalia K. Rojas[1], Emma Dalrymple[1], Shamez N. Ladhani[3,7‡], Roz Shafran[1‡], on behalf of the CLoCk Consortium[¶]

1 UCL Great Ormond Street Institute of Child Health, London, United Kingdom, 2 Division of Surgery & Interventional Science, Faculty of Medical Sciences, University College London, United Kingdom, 3 Immunisation Department, UK Health Security Agency, London, United Kingdom, 4 Department of Psychological Medicine, Institute of Psychiatry, Psychology and Neuroscience, King's College London, London, United Kingdom, 5 Department of Psychiatry, University of Cambridge, Cambridge, United Kingdom, 6 Centre for Medical Informatics, Usher Institute, University of Edinburgh, Edinburgh, United Kingdom, 7 Paediatric Infectious Diseases Research Group, St. George's University of London, London, United Kingdom

‡ SNL and RS are joint senior authors on this work.
¶ Membership of the CLoCk Consortium is provided in the Acknowledgments.
* snehal.pereira@ucl.ac.uk

## Abstract

### Background

Little is known about the prevalence and natural trajectory of post-COVID symptoms in young people, despite very high numbers of young people having acute COVID. To date, there has been no prospective follow-up to establish the pattern of symptoms over a 6-month time period.

### Methods

A non-hospitalised, national sample of 3,395 (1,737 SARS-COV-2 Negative;1,658 SARS-COV-2 Positive at baseline) children and young people (CYP) aged 11–17 completed questionnaires 3 and 6 months after PCR-confirmed SARS-CoV-2 infection between January and March 2021 and were compared with age, sex and geographically-matched test-negative CYP.

### Results

Three months after a positive SARS-CoV-2 PCR test, 11 of the 21 most common symptoms reported by >10% of CYP had reduced. There was a further decline at 6 months. By 3 and 6 months the prevalence of chills, fever, myalgia, cough and sore throat of CYP who tested

**Data Availability Statement:** Data is not publicly available. All requests for data will be reviewed by the Children & young people with Long Covid (CLoCk) study team, to verify whether the request is subject to any intellectual property or confidentiality obligations. Requests for access to the participant-level data from this study can be submitted via email to Clock@phe.gov.uk with detailed proposals for approval. A signed data access agreement with the CLoCk team is required before accessing shared data. Code is not made available as we have not used custom code or algorithms central to our conclusions.

**Funding:** Funding Funded by The Department of Health and Social Care, in their capacity as the National Institute for Health Research (NIHR), and by UK Research & Innovation (UKRI) who have awarded funding grant number COVLT0022. All research at Great Ormond Street Hospital NHS Foundation Trust and UCL Great Ormond Street Institute of Child Health is made possible by the NIHR Great Ormond Street Hospital Biomedical Research Centre. The views expressed are those of the authors and not necessarily those of the NHS, the NIHR, UKRI or the Department of Health. SMPP is supported by a UK Medical Research Council Career Development Award (ref: MR/P020372/1).

**Competing interests:** Terence Stephenson is Chair of the Health Research Authority and therefore recused himself from the Research Ethics Application. Trudie Chalder is a member of the National Institute for Health and Care Excellence committee for long COVID. She has written self-help books on chronic fatigue and has done workshops on chronic fatigue and post infectious syndromes. All remaining authors have no conflicts of interest. This does not alter our adherence to PLOS ONE policies on sharing data and materials.

positive for SARS-CoV-2 reduced from 10–25% at testing to <3%. The prevalence of loss of smell declined from 21% to 5% at 3 months and 4% at 6 months. Prevalence of shortness of breath and tiredness also declined, but at a lower rate. Among test-negatives, the same common symptoms and trends were observed at lower prevalence's. Importantly, in some instances (shortness of breath, tiredness) the overall prevalence of specific individual symptoms at 3 and 6 months was higher than at PCR-testing because these symptoms were reported in new cohorts of CYP who had not reported the specific individual symptom previously.

## Conclusions

In CYP, the prevalence of specific symptoms reported at time of PCR-testing declined with time. Similar patterns were observed among test-positives and test-negatives and new symptoms were reported six months post-test for both groups suggesting that symptoms are unlikely to exclusively be a specific consequence of SARS-COV-2 infection. Many CYP experienced unwanted symptoms that warrant investigation and potential intervention.

## Introduction

SARS-CoV-2 infection in children and young people (CYP) has generally been asymptomatic or a mild illness [1] compared to adults [2]. Of increasing concern, however, is long COVID (post-COVID-19 condition) which was first reported in adults and is of increasing concern in CYP [3]. We, like other NIHR/UKRI funded researchers use the term long COVID because this is used by the public, healthcare professionals and in searches and systematic reviews.

In our recent systematic literature review, we identified 21 eligible publications on long COVID in CYP [4]. The most common symptoms at 3 months were fatigue, insomnia, loss of smell, and headaches. Only five studies had a negative control group and only 13 studies reported the duration of follow-ups with longer follow-up associated with lower prevalence of cough, headache, cognitive problems and abdominal pain but higher prevalence of fever, fatigue, muscle pain, diarrhoea, loss of smell and difficulty breathing [4].

There is considerable variation in the published literature on the natural history of long COVID and which symptoms persist beyond 3 months in CYP [1, 5–10]. Single-centre follow-up studies may be biased towards those with sufficiently severe symptomatology to seek attendance at the hospital. Similarly, the much larger register- and health insurance-based studies may miss symptoms which do not reach the threshold for the parents or the child to seek healthcare follow-up.

The CLoCk study is the largest national, matched longitudinal cohort study of CYP in England [11], in which non-hospitalised CYP self-report post-COVID health problems after laboratory-confirmed SARS-CoV-2 infection compared to laboratory-confirmed SARS-CoV-2 test-negative CYP [11, 12]. Our original analysis of 6,084 participants [12] indicated that 35.4% who tested positive and 8.3% who tested negative were symptomatic at the time of testing, and that, 3 months after testing, 66.5% who tested positive and 53.3% who tested negative had any symptoms. The increase in symptoms from time of testing to 3 months post-test made it essential to extend our original analysis up to 6 months after PCR-testing in CYP to understand symptom trajectories. Our longitudinal cohort comprising SARS-CoV-2 PCR-positive and matched PCR-negative CYP allows us to describe (i) the natural course of baseline symptoms at 3 and 6 months after testing, (ii) the prevalence and course of symptoms reported for

the first time 3 months after testing, and (iii) those reporting symptoms for the first time 6 months after testing.

## Methods

The CLoCk study is described in detail elsewhere [11]. Briefly, CLoCk is a cohort study of SARS-CoV-2 PCR-positive CYP aged 11–17 years, matched by month of test, age, sex, and geographical area to SARS-CoV-2 test-negative CYP using the national SARS-CoV-2 testing dataset held by United Kingdom Health Security Agency (UKHSA).

The study aims to collect data on >30,000 CYP at 6, 12 and 24 months after a SARS-CoV-2 PCR test taken during September 2020-March 2021 [11]. Since participants were first contacted in April 2021, only the sub-sample of over 7,000 CYP respondents tested during January-March 2021 (Fig 1) could report, with minimal recall bias, symptoms at the time of testing (i.e., baseline data) and at ~3 months post-test. After obtaining online informed consent, CYP completed an online questionnaire about their health at the time of their PCR test ("baseline") and at the time of completing the questionnaire, approximately 3 months after their PCR test. A carer could assist younger CYP and CYP with special educational needs or disabilities. There were 6,804 participants in the data reported at 3 months (data collection date 18/06/21) [12]. Between 18/06/2021 and 16/08/21 an additional 292 participants filled in the 3-month questionnaire within 23 weeks from PCR testing and were therefore included in the target population for this analysis, making a total of 7,096 participants eligible to be included in the study. This group was approached again at 6 months post-test to complete their next online questionnaire. Of the CYP who answered both the 3- and 6-month questionnaires 141 of 1,984 CYP who originally tested PCR-negative, received a positive SARS-COV-2 test by 6 months and were excluded; similarly, 27 of the 1,758 who tested positive originally, were re-infected and excluded (determined by PCR tests results held by UKHSA and self-report of whether or not the CYP had a positive COVID-19 test). The final study sample comprised of 3,395 CYP (1,737 SARS-COV-2 Negative;1,658 SARS-COV-2 Positive at baseline, see Fig 1), representing a 47.8% response rate of participants eligible to be included (100%*3395/7,096).

In this final study sample, 287 of 1,737 PCR-negative and 257 of 1,658 PCR-positive CYP had received a COVID-19 vaccine between 3- and 6-months follow-up. Thirteen PCR-positive CYP attended hospital of whom 10 were admitted overnight.

## Measures

The questionnaire included demographics, elements of the International Severe Acute Respiratory and emerging Infection Consortium (ISARIC) Paediatric COVID-19 questionnaire [13], and the recent Mental Health of Children and Young people in England surveys [14]. Designed with ISARIC Paediatric Working Group as a harmonised data collection tool to facilitate international comparisons, it included 21 symptoms (mostly assessed as present/absent) [12] examined here. The questionnaire also included established scales for example the Strengths and Difficulties Questionnaire [15], Short Warwick Edinburgh Mental Wellbeing Scale [16], and Chalder Fatigue Scale [17], which will be reported separately. The questionnaire was largely unchanged between the 3- and 6-month follow-up except for removal of questions which were redundant at second contact: consent, pre-pandemic weight, postcode, ethnicity, symptoms, lifestyle and wellbeing at time of testing.

## Statistical methods

To assess representativeness of our study participants, we compared their demographic characteristics (sex, age, region of residence, and Index of Multiple Deprivation) to the target

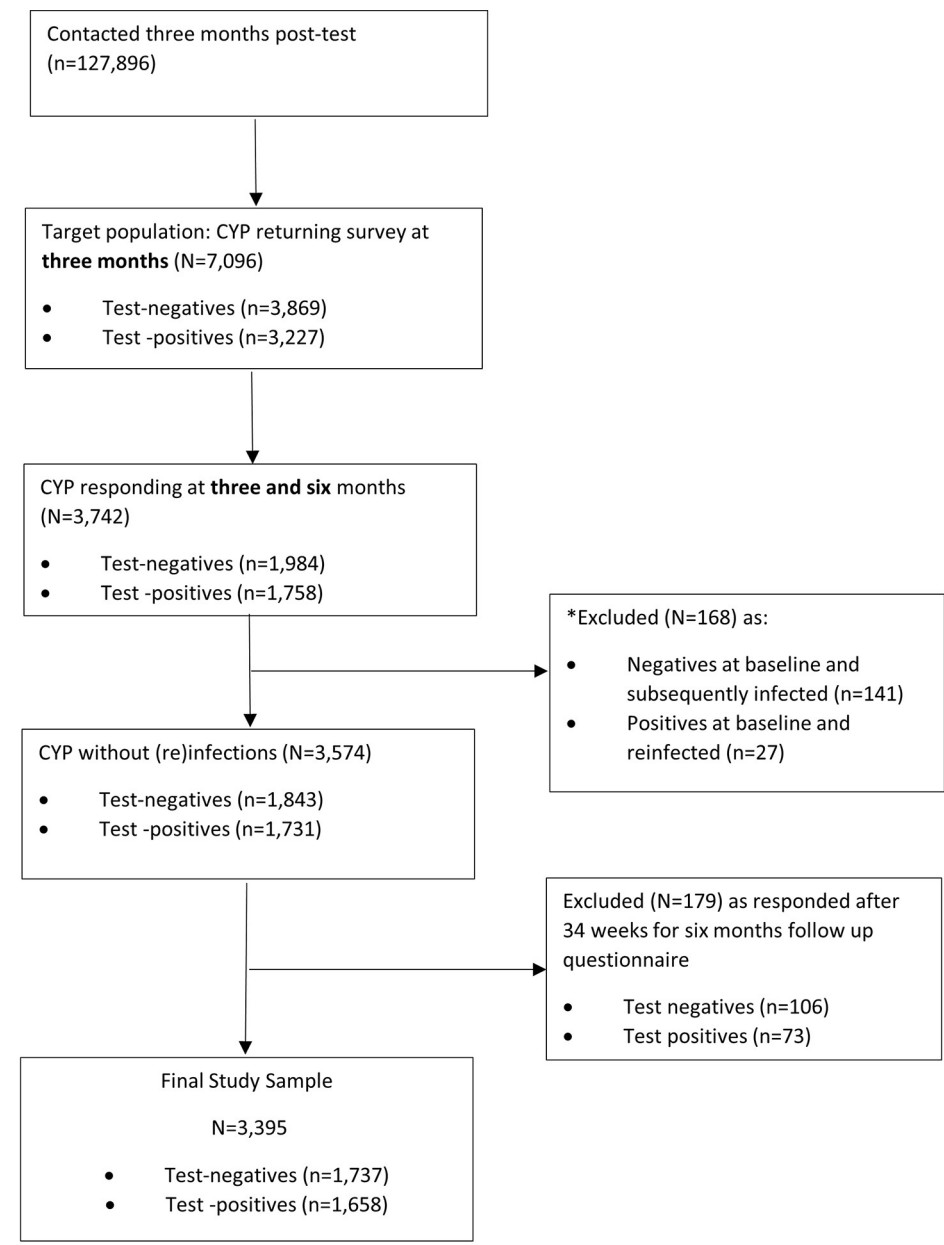

*(re)infections determined by PCR tests results held by UKHSA and self-report of whether or not the CYP had a positive COVID-19 test

**Fig 1. Participant flow diagram.**

population of all those who responded at 3 months post-test. We describe the prevalence of each of the 21 symptoms in two ways. First, we tabulate the prevalence (N (%)) of CYP who had a symptom never, once, twice or thrice (at testing, 3- and 6-months post-test) and assess whether the distribution of symptom prevalence differed by SARS-CoV-2 PCR status using a chi-squared test [18]. Second, we take into consideration the temporal nature of the data and generate stacked bar charts that show the distribution of symptoms across the three time-

points. We use stacked bar charts because they allow us to easily and visually demonstrate *when* each symptom was first reported. Both analyses were stratified by SARS-CoV-2 status. We inspected and grouped the bar charts into three broad categories as (i) symptoms where overall prevalence declined from baseline to 6 months post-test, (ii) symptoms where overall prevalence increased from baseline to 3 months and remained high at 6 months, and (iii) symptoms with relatively low prevalence (i.e., less than 10%) at any time point.

### Ethical approval and consent

Ethical approval was provided by the Yorkshire & The Humber—South Yorkshire Research Ethics Committee (REC reference: 21/YH/0060; IRAS project ID:293495). Public Health England (now UKHSA) has legal permission, provided by Regulation 3 of The Health Service (Control of Patient Information) Regulations 2002, to process patient confidential information for national surveillance of communicable diseases. Parents/carers were sent an invitation by post sent through PHE/UKHSA on behalf of the research team with a link to the website with the relevant information sheets and consent forms and they had the opportunity to ask any questions about the study. Parents/carers of CYP under 16 years of age were asked to complete an online parent/carer consent form. The young person was also asked to complete an online assent form to indicate their agreement. Consent was asked online from 16–17year-olds (using the Young Person Consent Form) in line with Health Research Authority recommended processes [19].

## Results

The 3 and 6-month follow-up questionnaires were returned at a median of 14.9 [IQR:13.1,18.9] and 27.4 [IQR: 25.9,28.6] weeks after testing, respectively. Overall, 1,737 of 3,869 (44.9%) of test-negative and 1,658 of 3,227 (51.4%) of test-positive CYP who responded at 3 months also responded at 6 months; and the analysed cohort was broadly similar to the target population (Table 1). In turn, our original analysis demonstrated that the respondents had similar demographics to the original invited cohort [12].

### Symptom profiles at baseline, 3 and 6-months post-test

The prevalence of CYP having a specific symptom never, once, twice or thrice is shown in Table 2. Among the test-positives 14.2% had at least one symptom at all three time points, while 15.7% never had any symptoms; corresponding values in the test-negatives were 1.8% and 35.0%. Thus, the distribution of symptom prevalence differed by SARS-CoV-2 PCR status (p<0.001, expect for experiencing sores/blisters on feet and 'other' symptoms). Among the test-positives, 35.9% had at least one symptom at testing, 67.8% had at least one symptom at 3 months and 56.6% had at least one symptom at 6 months. Corresponding values in the test-negatives were 9.2% (at testing) 53.3% (3 months post-test) and 35.3% (6 months post-test). We were able to categorise symptom patterns into three broad groups: (i) symptoms with overall prevalence increasing from baseline to 3 month and increasing or remaining high at 6 months (Fig 2), (ii) symptoms with overall prevalence declining from baseline to 6 months post-test (Fig 3) and (iii) symptoms with low overall prevalence at baseline (Fig 4).

 **In the test-positives**, of the more common 11 symptoms (Figs 2 and 3), when assessed by individual participant responses, reductions in the prevalence for all 11 specific individual symptoms were observed from baseline to 3 months' post-test, with a further, usually smaller, decline at 6 months. However, in some instances (shortness of breath, tiredness) the overall prevalence of specific individual symptoms at 3 and 6 months was higher because these symptoms were reported in a new cohort of CYP who had not reported that specific individual

**Table 1. Comparison of target population to analytic sample; and characteristics of children and young people in analytic sample by baseline PCR-test result: N (%).**

| Characteristic | | Target population of CYP who responded at 3 months post-test | CYP in analytic sample (responding at 3- and 6-months post-test) | SARS-CoV-2 Positive | SARS-CoV-2 Negative |
|---|---|---|---|---|---|
| **N** | | 7,096 | 3,395 | 1,658 | 1,737 |
| **SARS-CoV-2** | | | | | |
| | Negative | 3,869 (54.5) | 1,737 (51.2) | 0 (0.0) | 1,737 (100.0) |
| | Positive | 3,227 (45.5) | 1,658 (48.8) | 1,658 (100.0) | 0 (0.0) |
| **Age (years)** | | | | | |
| | 11–14 | 3,017 (42.5) | 1,346 (39.7) | 666 (40.2) | 680 (39.2) |
| | 15–17 | 4,079 (57.5) | 2,049 (60.3) | 992 (59.8) | 1,057 (60.8) |
| **Sex** | | | | | |
| | Male | 2,634 (37.1) | 1,178 (34.7) | 563 (34.0) | 615 (35.4) |
| | Female | 4,462 (62.9) | 2,217 (65.3) | 1,095 (66.0) | 1,122 (64.6) |
| **Ethnicity** | | | | | |
| | White | 5,251 (74.0) | 2,586 (76.2) | 1,242 (74.9) | 1,344 (77.4) |
| | Asian or Asian British | 1,056 (14.9) | 473 (13.9) | 249 (15.0) | 224 (12.9) |
| | Mixed | 354 (5.0) | 159 (4.7) | 76 (4.6) | 83 (4.8) |
| | Black, African, or Caribbean | 264 (3.7) | 110 (3.2) | 55 (3.3) | 55 (3.2) |
| | Other | 118 (1.7) | 47 (1.4) | 26 (1.6) | 21 (1.2) |
| | Prefer not to say | 53 (0.8) | 20 (0.6) | 10 (0.6) | 10 (0.6) |
| **IMD\*** | | | | | |
| | 1 (most deprived) | 1,466 (20.7) | 621 (18.3) | 311 (18.7) | 310 (17.8) |
| | 2 | 1,414 (19.9) | 688 (20.3) | 348 (21.0) | 340 (19.6) |
| | 3 | 1,359 (19.2) | 640 (18.9) | 300 (18.1) | 340 (19.6) |
| | 4 | 1,367 (19.3) | 683 (20.2) | 331 (20.0) | 352 (20.3) |
| | 5 (least deprived) | 1,490 (21.0) | 763 (22.5) | 368 (22.2) | 395 (22.7) |
| **Region** | | | | | |
| | East Midlands | 671 (9.5) | 342 (10.1) | 164 (9.9) | 178 (10.2) |
| | East of England | 1,127 (15.9) | 549 (16.2) | 245 (14.8) | 304 (17.5) |
| | London | 1,169 (16.5) | 542 (16.0) | 263 (15.8) | 279 (16.1) |
| | North East | 240 (3.4) | 95 (2.8) | 50 (3.0) | 45 (2.6) |
| | North West | 830 (11.7) | 367 (10.8) | 184 (11.1) | 183 (10.5) |
| | South East | 1,163 (16.4) | 605 (17.8) | 301 (18.2) | 304 (17.5) |
| | South West | 548 (7.7) | 280 (8.2) | 142 (8.5) | 138 (7.9) |
| | West Midlands | 853 (12.0) | 404 (11.9) | 203 (12.2) | 201 (11.6) |
| | Yorkshire and The Humber | 495 (7.0) | 211 (6.2) | 106 (6.4) | 105 (6.0) |

\*Index of Multiple Deprivation (IMD), derived from the CYP's lower super output area (a small local area level based geographic hierarchy), was used as a proxy for socio-economic status. We used IMD quintiles from most (quintile 1) to least (quintile 5) deprived.

symptom previously (Fig 2). In the PCR-positives, for example, the prevalence of unusual tiredness was 23.7% at baseline, 39.6% at 3-months and 40.9% at 6-months post-test. At 3 months, this comprised 12.5% who first reported being tired at baseline and 27.1% who were 'newly' tired at 3 months. At 6 months, there were 12.0% who first reported tiredness at baseline (which could further split into those who were tired at all three time points: 9.5% and who were tired at baseline and six months only: 2.5%), 19.0% who were tired at 3- and 6-months

**Table 2. Prevalence N (%) of CYP who had a symptom (never, once, twice or thrice) at testing, at 3 months post-test, at 6 months post-test.**

| SYMPTOM | SARS-CoV-2 PCR-Negative (N = 1,737) | | | | SARS-CoV-2 PCR-Positive (N = 1,658) | | | | |
|---|---|---|---|---|---|---|---|---|---|
| | Never had symptom | Only once | Only twice | Three times | Never had symptom | Only once | Only twice | Three times | p-value** |
| Fever | 1,624(93.5%) | 109(6.3%) | 4(0.2%) | 0(0.0%) | 1,336(80.6%) | 310 (18.7%) | 12(0.7%) | 0(0.0%) | <0.001 |
| Chills or shivers | 1,564(90.0%) | 141(8.1%) | 28(1.6%) | 4(0.2%) | 1,210(73.0%) | 360 (21.7%) | 76(4.6%) | 12(0.7%) | <0.001 |
| Headache | 1,327(76.4%) | 315 (18.1%) | 88(5.1%) | 7(0.4%) | 846(51.0%) | 531 (32.0%) | 223 (13.4%) | 58(3.5%) | <0.001 |
| Loss of smell/taste | 1,670(96.1%) | 58(3.3%) | 7(0.4%) | 2(0.1%) | 1,143(68.9%) | 349 (21.0%) | 117(7.1%) | 49(3.0%) | <0.001 |
| Unusual strong muscle pains | 1,658(95.5%) | 67(3.9%) | 12(0.7%) | 0(0.0%) | 1,367(82.4%) | 235 (14.2%) | 48(2.9%) | 8(0.5%) | <0.001 |
| Persistent cough | 1,597(91.9%) | 117(6.7%) | 20(1.2%) | 3(0.2%) | 1,329(80.2%) | 292 (17.6%) | 33(2.0%) | 4(0.2%) | <0.001 |
| Sore throat | 1,488(85.7%) | 215 (12.4%) | 33(1.9%) | 1(0.1%) | 1,116(67.3%) | 470 (28.3%) | 69(4.2%) | 3(0.2%) | <0.001 |
| Skipping meals | 1,530(88.1%) | 160(9.2%) | 46(2.6%) | 1(0.1%) | 1,257(75.8%) | 286 (17.2%) | 100(6.0%) | 15(0.9%) | <0.001 |
| Unusual shortness of breath | 1,455(83.8%) | 196 (11.3%) | 79(4.5%) | 7(0.4%) | 1,069(64.5%) | 293 (17.7%) | 232 (14.0%) | 64(3.9%) | <0.001 |
| Unusual Fatigue/tiredness | 1,101(63.4%) | 373 (21.5%) | 243 (14.0%) | 20(1.2%) | 652(39.3%) | 443 (26.7%) | 405 (24.4%) | 158(9.5%) | <0.001 |
| Dizziness or light-headedness | 1,481(85.3%) | 188 (10.8%) | 64(3.7%) | 4(0.2%) | 1,129(68.1%) | 377 (22.7%) | 129(7.8%) | 23(1.4%) | <0.001 |
| Unusually hoarse voice | 1,679(96.7%) | 53(3.1%) | 5(0.3%) | 0(0.0%) | 1,535(92.6%) | 111(6.7%) | 11(0.7%) | 1(0.1%) | <0.001 |
| Unusual chest pain/tightness in chest | 1,629(93.8%) | 90(5.2%) | 17(1.0%) | 1(0.1%) | 1,357(81.8%) | 233 (14.1%) | 58(3.5%) | 10(0.6%) | <0.001 |
| Unusual abdominal pain | 1,627(93.7%) | 90(5.2%) | 16(0.9%) | 4(0.2%) | 1,483(89.4%) | 146(8.8%) | 25(1.5%) | 4(0.2%) | <0.001 |
| Diarrhoea | 1,655(95.3%) | 67(3.9%) | 14(0.8%) | 1(0.1%) | 1,501(90.5%) | 132(8.0%) | 23(1.4%) | 2(0.1%) | <0.001 |
| Confusion, disorientation, or drowsiness | 1,622(93.4%) | 89(5.1%) | 25(1.4%) | 1(0.1%) | 1,412(85.2%) | 182 (11.0%) | 52(3.1%) | 12(0.7%) | <0.001 |
| Unusual eye-soreness or discomfort | 1,625(93.6%) | 96(5.5%) | 16(0.9%) | 0(0.0%) | 1,381(83.3%) | 222 (13.4%) | 48(2.9%) | 7(0.4%) | <0.001 |
| Earache or ringing in your ears | 1,587(91.4%) | 122(7.0%) | 25(1.4%) | 3(0.2%) | 1,412(85.2%) | 202 (12.2%) | 40(2.4%) | 4(0.2%) | <0.001 |
| Raised, red, itchy welts on the skin | 1,712(98.6%) | 25(1.4%) | 0(0.0%) | 0(0.0%) | 1,600(96.5%) | 50(3.0%) | 8(0.5%) | 0(0.0%) | <0.001 |
| Red/purple sores or blisters on your feet | 1,704(98.1%) | 28(1.6%) | 4(0.2%) | 1(0.1%) | 1,624(97.9%) | 28(1.7%) | 6(0.4%) | 0(0.0%) | 0.686 |
| Other | 1,430(82.3%) | 289 (16.6%) | 17(1.0%) | 1(0.1%) | 1,386(83.6%) | 246 (14.8%) | 24(1.4%) | 2(0.1%) | 0.280 |
| At least one symptom* | 608(35.0%) | 731 (42.1%) | 367 (21.1%) | 31 (1.8%) | 260 (15.7%) | 589 (35.5%) | 573 (34.6%) | 236 (14.2%) | <0.001 |

*at least one of the 21 listed symptoms

**p-value from chi-squared test

post-test (but not at baseline) and 9.9% who were 'newly' tired at 6 months. The prevalence of dizziness remained relatively constant at baseline, 3- and 6-months (Fig 2).

For the remaining eight of the more common 11 symptoms (Fever, Chills, Headache, Loss of smell, Muscle pain, Persistent cough, Sore throat, Skipping meals), the overall prevalence of the specific individual symptoms declined from baseline to 6 months post-test because the proportion reporting new onset for these symptoms was relatively low (Fig 3).

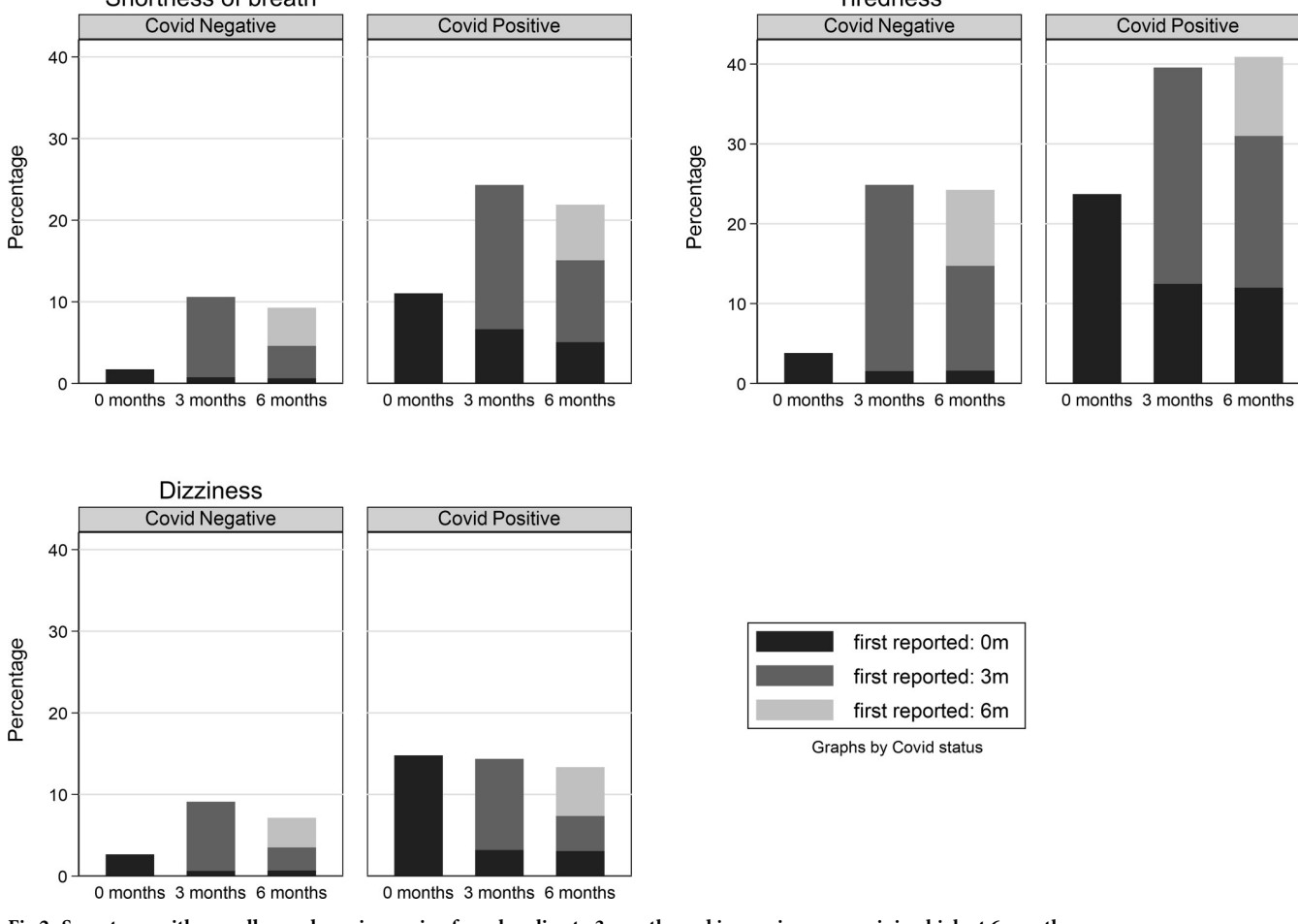

**Fig 2. Symptoms with overall prevalence increasing from baseline to 3 months and increasing or remaining high at 6 months.**

The prevalence of the other ten of the 21 symptoms we enquired about was less than 10% at baseline (Fig 4 –note the much smaller vertical scale); for 9 of these symptoms, the prevalence at any time point (baseline, 3- and 6-months post-test) was always less than 10%. The symptom 'other' just exceeded 10% at 3-months post-test.

**For those who tested negative at baseline**, the prevalence of two symptoms (shortness of breath and unusual tiredness) which occurred in more than 10% of test negative respondents at some point, increased from baseline to 3 months and remained stable at 6 months (Fig 2). This is because, similar to the test-positives, reductions at 3- and 6-months among CYP reporting those symptoms at baseline were offset by a new cohort of CYP who first reported these symptoms at 3 months. Similarly, while the prevalence in CYP reporting the individual specific symptoms for the first time at 3 months post-test declined by 6 months, a new cohort of CYP reported that same specific individual symptom at 6 months post-test. At 6 months, the difference in prevalence between test-positives and test-negatives varied by *when* the symptom was first reported: for example, the difference in prevalence between test-positives and test-negatives reporting shortness of breath for the first time at baseline and also experiencing it at 6 months was 4.43% (95% CI:3.31%,5.55%); the difference in prevalence among those reporting shortness of breath for the first time at 6 months was 2.15% (95% CI:0.58%,3.72%), see Table 3.

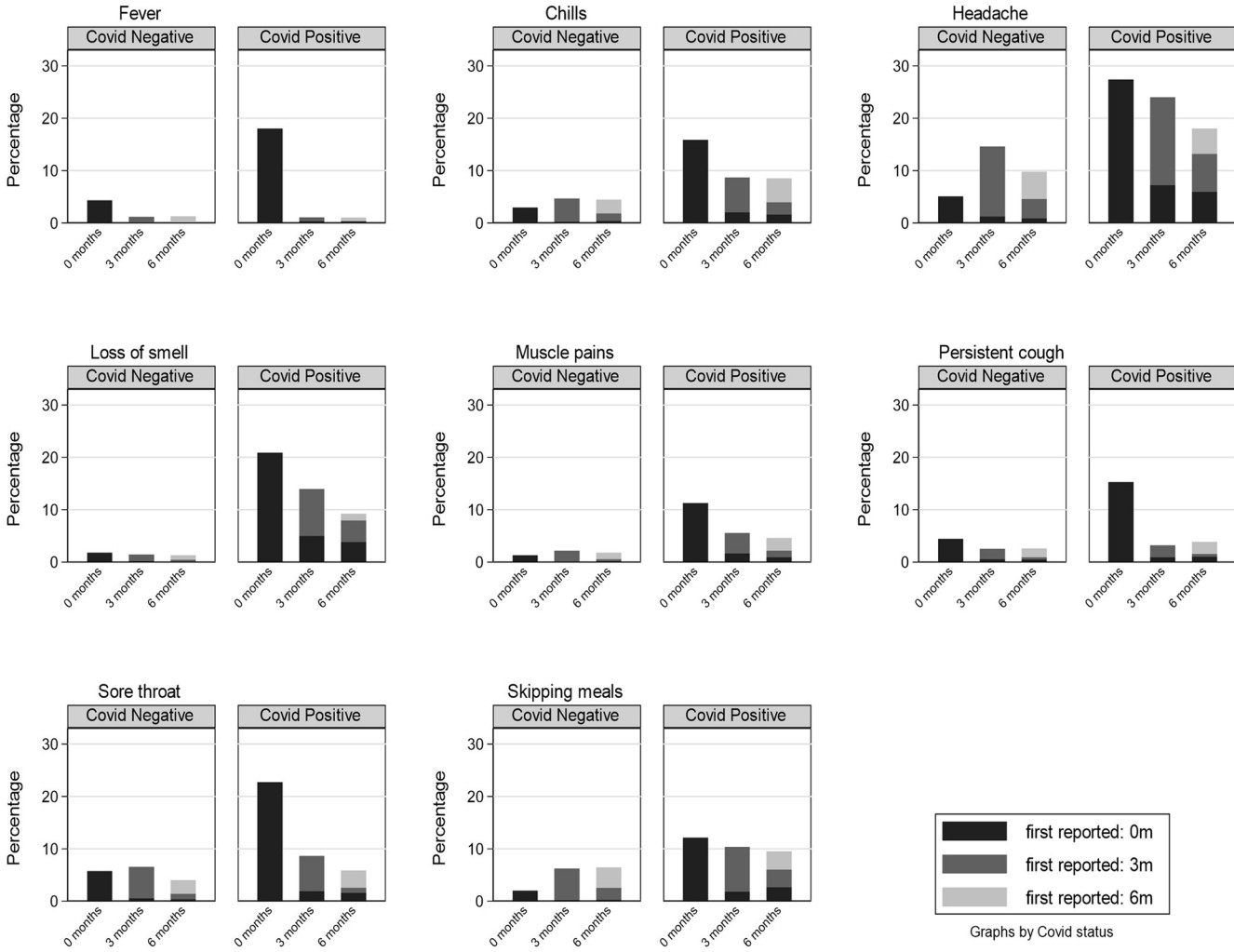

*For all symptoms (except skipping meals) p-value for difference between proportion of CYP with symptom at baseline and 6 months post-test, in test-positives was <0.001; for skipping meals p-value=0.02.

**Fig 3.** Symptoms with overall prevalence declining from baseline to 6 months post-test in test-positives*.

The prevalence of two symptoms (headache and other), which occurred in more than 10% of test-negative respondents at some point, increased at 3 months and then fell at 6 months (Figs 3 and 4). Again, many of the test negatives who described these two symptoms at 3 months were no longer experiencing them at 6 months but some young people first describe the symptom at 6 months.

The remaining 17 of the 21 symptoms were present in less than 10% of CYP at any of the 3 time points and show a flat pattern over time (Figs 2–4).

## Discussion

The aim of this paper was to describe the prevalence of symptoms in a large, national, longitudinal cohort of PCR-positive and PCR-negative CYP at the time of PCR testing and after 3 and 6 months. The results show that at all three time points, symptoms were more common in

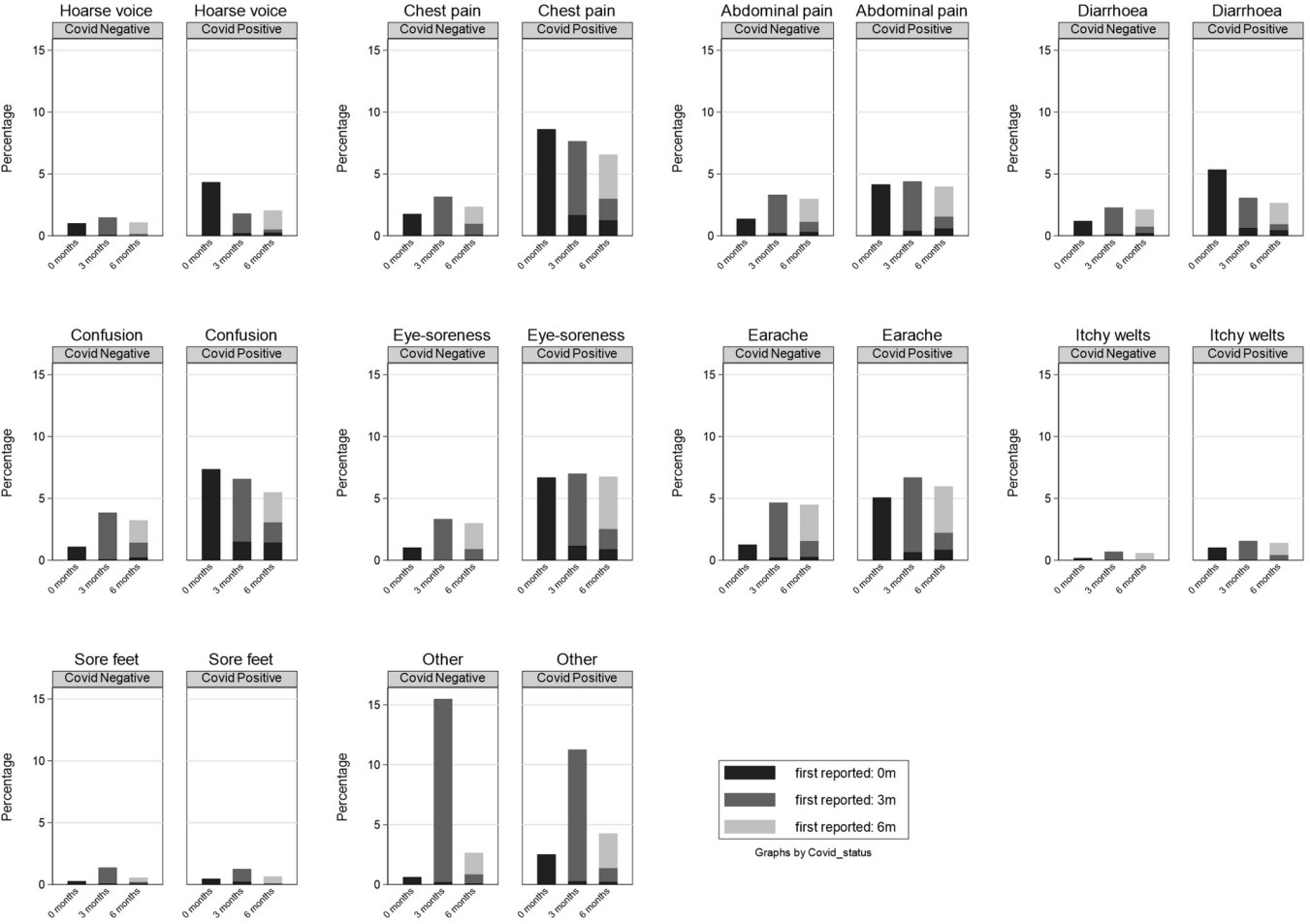

**Fig 4. Symptoms with low overall prevalence (less than 10%) at baseline.**

**Table 3. Difference in prevalence of shortness of breath, tiredness and dizziness at 6 months post-test, between test-positives and test-negatives, by time symptom first reported.**

| | Prevalence difference at 6-months post-test between test-positives and test-negatives (95% CI) |
|---|---|
| Shortness of breath | |
| first reported at: | |
| 0 months | 4.43 (3.31, 5.55) |
| 3 months | 6.04 (4.33, 7.75) |
| 6 months | 2.15 (0.58, 3.72) |
| Tiredness | |
| first reported at: | |
| 0 months | 10.39 (8.72, 12.06) |
| 3 months | 5.87 (3.40, 8.34) |
| 6 months | 0.39 (-1.60, 2.38) |
| Dizziness | |
| first reported at: | |
| 0 months | 2.38 (1.47, 3.30) |
| 3 months | 1.46 (0.21, 2.71) |
| 6 months | 2.34 (0.90, 3.78) |

test-positive compared to test-negative CYP. Many of the test-positive CYP who had a particular symptom at testing were free from that symptom at both 3- and 6-months post-test, suggesting that those symptoms were on a downward trend with time. Additionally, most CYP who first developed a particular symptom at 3 months after their positive PCR-test did not report that symptom at 6 months. Indeed, for all 21 symptoms and for both PCR positive and negative CYP, we found that different cohorts developed individual symptoms at baseline, 3- and 6-months post-test, but these symptoms resolved upon questioning at the next time point. However, it is also notable that at 6 months the data show that some young people were describing symptoms for the first time. These findings have not previously been reported.

We were able to identify three broad groups of symptom patterns: symptoms with the overall prevalence increasing from baseline to 3 months and increasing or remaining high at 6 months; symptoms with the overall prevalence declining from baseline to 6-months post-test; and symptoms with low overall prevalence at baseline. Of these three patterns, the commonest symptoms followed a pattern typical of a viral upper respiratory infection or winter 'flu in teenagers (Fig 3). The symptoms of chills, fever, muscle pain, cough and sore throat affected a tenth to a quarter of CYP at the time of their positive SARS-COV-2 test but, by 3 and 6 months, these symptoms persisted in less than 3%. Loss of smell, which is almost unique to acute SARS-COV-2 infection, persisted in 5% at 3 months and 4% at 6 months. In the second pattern (Fig 2), a substantial proportion of responders who reported not suffering from shortness of breath or tiredness at the time of their PCR-positive and PCR-negative test had developed these two symptoms by 3 months, with some CYP first reporting these symptoms at 6 months post-test.

With the exception of eye soreness, the proportion of CYP developing any of the other 20 symptoms anew between 3- and 6-months post-test was remarkably similar for test-positives and test-negatives. The reporting of new symptoms six months post-test for both groups suggests that these are not new long COVID symptoms arising in the PCR-positive CYP as an exclusive consequence of the initial SARS-COV-2 infection. The prevalence of miscellaneous, non-specific 'other symptoms' in both test-positive and test-negative groups was very similar (Fig 4), consistent with these being unrelated to SARS-COV-2 infection. New symptoms developing 3 months or more after infection and in both test positives and test negatives may represent the background level of new symptomatology in teenagers in England during these months. This emphasises the need for control groups in long COVID studies and benchmarking against rates of symptoms in the general population before, during and after the pandemic.

In adult studies of persisting post-COVID symptoms, symptom prevalence also declines with time since infection albeit at a variable rate depending on study method, setting and population. In a review of symptoms persisting for 3- to 6-months after testing (mean age varied across studies from 38 to 63 years), prevalence of persisting symptoms was lower (range 2.3–62%, median 26%) in studies that predominantly included non-hospitalised patients [6]. Nine observational cohort studies, one cross-sectional study and one case-control study allowed assessment of long COVID prevalence at more than 6 months of follow-up. In the six studies with mainly non-hospitalised or exclusively ambulatory patients, the prevalence ranged between 12.8–53% (median, 25%).

Such data in CYP are more limited and inconsistent. Some studies indicate a steady decline in symptoms. For example, in a single-centre cohort of 171 SARS-CoV-2 positive children (median age, 3 [IQR 1–8] years) from Australia, all children had returned to their baseline health status when followed-up in March 2021 (1–1.5 years post-infection) [10]. Similarly, a national Danish cohort study of 37,522 0–17 year-olds with SARS-CoV-2 infection verified by PCR and a control group of 78,037 randomly selected children, who had not tested positive for SARS-CoV-2 [5], reported that in most children, 'long COVID' symptoms resolved within 1–5

months. However, this analysis has been criticised for failing to account for the differing observation periods [20].

In contrast, other studies indicate continued difficulties. A single centre prospective cohort study of children (≤18 years old) admitted with suspected or confirmed Covid-19 in Moscow, Russia, followed up 518 of 853 (61%) eligible children after hospital discharge [9]. Their median age was 10.4 years (IQR, 3–15.2). At follow-up interview, 126 (24.3%) participants reported persistent symptoms at a median of 256 days post-discharge, among which fatigue (n = 53, 10.7%), sleep disturbance (n = 36, 6.9%,) and sensory problems (29, 5.6%) were the most common. Multiple symptoms were experienced by 44 (8.4%) participants. Persistent symptoms were more common among older children. Using a different methodology, a larger population study from Norway used nationwide register data to estimate the impact of COVID-19 on long-term healthcare use among 706,855 CYP [8]. The authors identified a short-term increase in primary (but not specialist) care use after COVID-19 related to respiratory and general or non-specific conditions, mostly in the four weeks after infection. The increase in primary care use persisted for up to six months among 1–5-year-olds.

The conflicting findings from previous studies emphasise the importance of a 'within individual' longitudinal study design, as highlighted by the substantial heterogeneity across symptoms and time and no previous study looked at pattern of individual symptoms across time in the way that has been done in the current analysis. If we had simply looked at cross-sectional prevalence of symptoms at baseline, 3 months and 6 months, the variability in symptom patterns would not have emerged, nor would the finding that test-positive and test-negative CYP are reporting new symptoms at 6 months. Rather, it would appear as if the prevalence's of the commonest post-COVID symptoms stayed largely stable over time. It is our view that new symptoms should not be exclusively viewed as new long COVID symptoms arising as a consequence of the initial SARS-COV-2 infection but instead should be seen in the wider context of prevalence of such symptoms in the general population. Data on such symptom prevalence in the general population is surprisingly difficult to obtain but is essential to understand the trajectory of symptoms specifically attributable to SARS-COV-2 infection. However, regardless of whether the symptoms are or are not attributable to SARS-COV-2 infection, it is clear that there are many young people experiencing a range of unwanted and impairing symptoms that warrant clinical investigation and potential intervention.

The CLoCk study has limitations which have been discussed [11, 12], and here we detail main limitations in relation to the current manuscript. In particular, the response rate for the 6-month follow-up questionnaire completion was 47.8% (3,395 of 7,096) but there was little difference in demographic characteristics between test-positive and test-negative participants, reflecting the matched-cohort study design. While we excluded CYP who had been (re) infected after their baseline PCR-test, we acknowledge that (re)infections may have gone undetected. However, the very low prevalence of loss of smell/taste among test-negatives, at testing 3- and 6-months post-test, provides some reassurance of a low rate of unconfirmed SARS-CoV-2 infections in the test-negatives. Symptoms at baseline are subject to recall bias as they were reported 3-months post-test; however, 3-month and 6-month symptoms were reported prospectively. We made the objective decision to describe all 21 symptoms in our manuscript, rather than using data reduction strategies to derive the underlying structure of symptoms. It is our view that understanding the impact of the individual symptoms as well as their collective impact is required to fully understand impairment resulting from SARS-COV-2 infection; in this manuscript we chose to focus on the former. The explanation for the reporting of new symptoms at 6 months warrants further exploration and context. However, the study points to the importance of longitudinally tracking symptoms in the same individuals over time alongside matched test-negatives to properly understand the long-term impact of COVID infection in CYP.

## Conclusions

In conclusion, while acknowledging that (re)infections may have gone undetected, our study demonstrates why new symptoms arising 6-months post-infection should not be exclusively viewed as new long COVID symptoms and should be seen within the wider context of symptomology in the general population. Importantly, regardless of whether symptoms are (or are not) attributable to SARS-COV-2 infection, many CYP experiencing unwanted symptoms that warrant investigation and potential intervention.

## Acknowledgments

Michael Lattimore, UKHSA, as Project Officer for the CLoCk study.

Olivia Swann and Elizabeth Whittaker designed the elements of the ISARIC Paediatric COVID-19 follow-up questionnaire which were incorporated into the online questionnaire used in this study to which all the CLoCk Consortium members contributed.

## Additional Co-Applicants on the grant application and CLoCk Consortium members (alphabetical)

Marta Buszewicz, University College London, m.buszewicz@ucl.ac.uk
Esther Crawley, University of Bristol, Esther.Crawley@bristol.ac.uk
Bianca De Stavola, University College London, b.destavola@ucl.ac.uk
Shruti Garg, University of Manchester, Shruti.Garg@mft.nhs.uk
Dougal Hargreaves, Imperial College London, d.hargreaves@imperial.ac.uk
Anthony Harnden, Oxford University, anthony.harnden@phc.ox.ac.uk
Michael Levin, Imperial College London, m.levin@imperial.ac.uk
Vanessa Poustie, University of Liverpool, v.poustie@liverpool.ac.uk
Terry Segal, University College London Hospitals NHS Foundation Trust, terry.segal@nhs.net
Malcolm Semple, University of Liverpool, M.G.Semple@liverpool.ac.uk
Kishan Sharma, Manchester University NHS Foundation Trust (sadly deceased)
Elizabeth Whittaker, Imperial College London, e.whittaker@imperial.ac.uk

## Author Contributions

**Conceptualization:** Terence Stephenson, Roz Shafran.

**Data curation:** Snehal M. Pinto Pereira, Manjula D. Nugawela, Kelsey McOwat, Ruth Simmons.

**Formal analysis:** Snehal M. Pinto Pereira, Manjula D. Nugawela.

**Funding acquisition:** Terence Stephenson, Trudie Chalder, Tamsin Ford, Isobel Heyman, Olivia V. Swann, Emma Dalrymple, Roz Shafran.

**Methodology:** Terence Stephenson, Trudie Chalder, Tamsin Ford, Isobel Heyman, Olivia V. Swann, Emma Dalrymple, Shamez N. Ladhani, Roz Shafran.

**Project administration:** Kelsey McOwat, Roz Shafran.

**Resources:** Kelsey McOwat, Shamez N. Ladhani.

**Writing – original draft:** Terence Stephenson, Snehal M. Pinto Pereira, Lana Fox-Smith, Natalia K. Rojas.

**Writing – review & editing:** Kelsey McOwat, Ruth Simmons, Trudie Chalder, Tamsin Ford, Isobel Heyman, Olivia V. Swann, Lana Fox-Smith, Natalia K. Rojas, Emma Dalrymple, Shamez N. Ladhani, Roz Shafran.

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
