## [Decision Letter · Decision Letter 0]

24 Aug 2022

PONE-D-22-21048Long COVID - Six months of prospective follow-up of changes in symptom profiles of non-hospitalised children and young people after SARS-CoV-2 testing: a national matched cohort study (The CLoCk) Study.PLOS ONE

Dear Dr. Pinto Pereira,

Thank you for submitting your manuscript to PLOS ONE. After careful consideration, we feel that it has merit but does not fully meet PLOS ONE’s publication criteria as it currently stands. Therefore, we invite you to submit a revised version of the manuscript that addresses the points raised during the review process.

We look forward to receiving your revised manuscript.

Kind regards,

Dong Keon Yon, MD, FACAAI

Academic Editor

PLOS ONE

Journal Requirements:

3. You indicated that you had ethical approval for your study. In your Methods section, please ensure you have also stated whether you obtained consent from parents or guardians of the minors included in the study or whether the research ethics committee or IRB specifically waived the need for their consent.

"Terence Stephenson is Chair of the Health Research Authority and therefore recused himself from the Research Ethics Application. Trudie Chalder is a member of the National Institute for Health and Care Excellence committee for long COVID. She has written self-help books on chronic fatigue and has done workshops on chronic fatigue and post infectious syndromes. All remaining authors have no conflicts of interest. "

Additional Editor Comments:

Thank you for submitting your manuscript. The reviewers and I believe it is of potential value for our readers. However, the reviewers have raised a number of very important issues, and their excellent comments will need to be adequately addressed in a revision before the acceptability of your manuscript for publication in the Journal can be determined. We cannot guarantee that your revised paper will be chosen for publication; this would be solely based on how satisfactorily you have addressed the reviewer comments.

Reviewers' comments:

Reviewer's Responses to Questions

**Comments to the Author**

1. Is the manuscript technically sound, and do the data support the conclusions?

Reviewer #1: Partly

Reviewer #2: Yes

Reviewer #3: Yes

2. Has the statistical analysis been performed appropriately and rigorously? 

Reviewer #1: No

Reviewer #2: I Don't Know

Reviewer #3: Yes

3. Have the authors made all data underlying the findings in their manuscript fully available?

Reviewer #1: No

Reviewer #2: Yes

Reviewer #3: Yes

4. Is the manuscript presented in an intelligible fashion and written in standard English?

Reviewer #1: Yes

Reviewer #2: Yes

Reviewer #3: Yes

5. Review Comments to the Author

Reviewer #1: This is an interesting paper from an important study, elements of which have been described elsewhere. It is a large study. The response medium was not self-evident from the paper, but I presume was electronic rather than paper-based. For a contemporary follow-up study, the follow up rate was pretty reasonable, but Professor Dillman would be deeply disappointed such an outcome- it is always a challenge in such studies to balance overall numbers of respondents with the persistence needed to get a higher rate of return from a smaller group. My main issue however is with the analysis.

The sample size is large- it is not clear to my why univariate analyses have need used, rather than data reduction strategies such as principle component and factor analyses to get at the underlying structure of symptomatic responses. The authors argue that that the late-onset' symptoms probably represents a late onset of a 'non-specific viral illness. This would be amuch more powerful argument if an initial cluster has been identified at an earlier point, with more COVID specific symptoms forming other components. The authors include at least two mentions of authors with statistical expertise-a fuller discussion of the analytic approach adopted here would be valuable.

And I would also add that the 'stacked bar graphs are, as often the case, very hard to interpret. Line graphs which allow an easy interpretation of the distribution of symptoms across time are much easier to follow, and would make the emergence of a 'late onset symptom population umore obvious.

As a small point, the issue of exclusion of subjects who reported testing positive during teh study period should be clearly identified in the text, as its an obvious issue.

Reviewer #2: Thank you for the opportunity to review this research article.

This is a very timely and interesting study. I would like to complement authors for maintaining rigour in the conduct of this study and for providing adequate details to convey their findings. I did not find any major issues in the conduct and reporting of the study. However, I would just make a very minor suggestion – Authors have mentioned CLoCK study on several occasions and have cited previous findings from this study, however I think that the current manuscript should stand on its own and readers should have enough information within current manuscript without referring to other publications. So, maybe authors can provide additional details about CLoCK study within current manuscript.

In the Method section, mention the response rate in the recruitment process.

In figure 1. The word “reinfected (n-27)” is covered by another box. Please fix it.

Another suggestion would be to have a “Conclusion” section at the end. In the current format, the last paragraph consist of limitations and importance of this study. I would recommend adding the conclusion section and highlighting some of the key findings. The statements in line 351 on page 15 “It is our view that new symptoms…….” And line 355 on page 15 “ However, regardless of whether symptoms…..” are very important statements, so in my opinion they should be part of conclusion section and should be highlighted in the abstract too.

Reviewer #3: Thank you for inviting me as a reviewer of this article.

I would like to suggest adding following sentence and citation on the "Statistical methods" section.

Prevalence N (%) of CYP who had a symptom (never, once, twice or thrice) at testing, at 3 months 211 post-test, at 6 months post-test was tested by Chi-squared test [1]

[1] https://doi.org/10.54724/lc.2022.e1

6. PLOS authors have the option to publish the peer review history of their article (what does this mean?). If published, this will include your full peer review and any attached files.

Reviewer #1: No

Reviewer #2: No

Reviewer #3: No

---

## [Author Response · Author response to Decision Letter 0]

18 Oct 2022

Dear Dr. Yon,

Thank you for sending our manuscript out for review and providing the opportunity to submit a revised version. We have responded to all the comments, with appropriate edits to the manuscript as indicated below. We believe that the comments and suggestions have strengthened the paper and hope that the manuscript is now suitable for publication in PLoS One.

As requested, I confirm our updated competing interests statement:

“Terence Stephenson is Chair of the Health Research Authority and therefore recused himself from the Research Ethics Application. Trudie Chalder is a member of the National Institute for Health and Care Excellence committee for long COVID. She has written self-help books on chronic fatigue and has done workshops on chronic fatigue and post infectious syndromes. All remaining authors have no conflicts of interest. This does not alter our adherence to PLOS ONE policies on sharing data and materials.”

Snehal Pinto Pereira

(on behalf of all authors)

Reviewer #1: This is an interesting paper from an important study, elements of which have been described elsewhere. It is a large study. The response medium was not self-evident from the paper, but I presume was electronic rather than paper-based. 

Response: Thank you for acknowledging that our manuscript, based on a large important study with a reasonable follow-up rate, is interesting. The reviewer is correct that the response medium was electronic (see details in 2nd paragraph of Methods: “After obtaining informed consent, CYP completed an online questionnaire about their health …”). 

The sample size is large- it is not clear to my why univariate analyses have need used, rather than data reduction strategies such as principle component and factor analyses to get at the underlying structure of symptomatic responses. The authors argue that that the late-onset' symptoms probably represents a late onset of a 'non-specific viral illness. This would be a much more powerful argument if an initial cluster has been identified at an earlier point, with more COVID specific symptoms forming other components. The authors include at least two mentions of authors with statistical expertise-a fuller discussion of the analytic approach adopted here would be valuable.

Response: The aim of the study was to describe the “prevalence and natural trajectory of post-COVID symptoms in young people” (see Abstract). Under “Measures” we explain that the 21 symptoms we examine in the manuscript were determined with the International Severe Acute Respiratory and emerging Infection Consortium (ISARIC) Paediatric Working Group. We made the objective decision to describe all 21 symptoms in our manuscript because they will serve as a useful comparison to other studies that use ISARIC’s COVID-19 questionnaire. Moreover, the Delphi consensus research definition of Long COVID (Archives of Disease in Childhood 2022;107:674-680) focuses on having at least 1 symptom (as a single symptom can be impairing). Given the potential heterogeneity of experience due to each symptom, each one is worthy of consideration in its own right. Therefore, data reduction strategies (results of which could vary depending on the specific dataset examined) were not employed. We now acknowledge these points in the discussion (lines 422-427): “We made the objective decision to describe all 21 symptoms in our manuscript, rather than using data reduction strategies to derive the underlying structure of symptoms. It is our view that understanding the impact of the individual symptoms as well as their collective impact is required to fully understand impairment resulting from SARS-COV-2 infection; in this manuscript we chose to focus on the former.” We have also edited the “Statistical methods” (lines 177-189) to include a fuller discussion of the analytic approach as requested taking in both this reviewer’s comments and those of reviewer #3 it now reads: “To assess representativeness of our study participants, we compared their demographic characteristics (sex, age, region of residence, and Index of Multiple Deprivation) to the target population of all those who responded at 3 months post-test. We describe the prevalence of each of the 21 symptoms in two ways. First, we tabulate the prevalence (N (%)) of CYP who had a symptom never, once, twice or thrice (at testing, 3- and 6-months post-test) and assess whether the distribution of symptom prevalence differed by SARS-CoV-2 PCR status using a chi-squared test [18]. Second, we take into consideration the temporal nature of the data and generate stacked bar charts that show the distribution of symptoms across the three time-points. We use stacked bar charts because they allow us to easily and visually demonstrate when each symptom was first reported. Both analyses were stratified by SARS-CoV-2 status. We inspected and grouped the bar charts into three broad categories as (i) symptoms where overall prevalence declined from baseline to 6 months post-test, (ii) symptoms where overall prevalence increased from baseline to 3 months and remained high at 6 months, and (iii) symptoms with relatively low prevalence (i.e., less than 10%) at any time point.”

And I would also add that the 'stacked bar graphs are, as often the case, very hard to interpret. Line graphs which allow an easy interpretation of the distribution of symptoms across time are much easier to follow, and would make the emergence of a 'late onset symptom population more obvious.

Response: Thank you for this comment. We too were concerned about how best to present the data for easy interpretation of the distribution of symptoms across time. In our initial exploration we drew both line graphs as well as stacked bar graphs (see example below for tiredness in test-positives). We decided that the stacked bar charts allow us to emphasise when each symptom was first reported better than the line graphs. Moreover, we thought that joining our data points together, would be misleading as it suggests that we know what the prevalence is between our data collection time points, when in fact we do not. Therefore, we decided that stacked bar charts were more appropriate. We have presented these data several times to both clinical and statistical colleagues as well as our Patient and Public Involvement group and the feedback we have had has universally been positive. Rather than finding stacked bar graphs being very hard to interpret, audiences have found them immediately intuitive and compelling. The feedback we have had is that they can immediately appreciate that many of the young people first reporting a symptom at 3 months did not have that symptom when tested and many of the young people first reporting a symptom at 6 months did not have that symptom at 3 months. However, if the editor would like us to redraw all graphs using line graphs we can do so.

As a small point, the issue of exclusion of subjects who reported testing positive during the study period should be clearly identified in the text, as it is an obvious issue.

Response: We thank the reviewer for this important point and have added text to lines 142-150, which now reads: “Of the CYP who answered both the 3- and 6-month questionnaires 141 of 1,984 CYP who originally tested PCR-negative, received a positive SARS-COV-2 test by 6 months and were excluded; similarly, 27 of the 1,758 who tested positive originally, were re-infected and excluded (determined by PCR tests results held by UKHSA and self-report of whether or not the CYP had a positive COVID-19 test).” We have also added this information to a footnote to Figure 1.

Reviewer #2

Thank you for the opportunity to review this research article. This is a very timely and interesting study. I would like to complement authors for maintaining rigour in the conduct of this study and for providing adequate details to convey their findings. I did not find any major issues in the conduct and reporting of the study. However, I would just make a very minor suggestion – Authors have mentioned CLoCK study on several occasions and have cited previous findings from this study, however I think that the current manuscript should stand on its own and readers should have enough information within current manuscript without referring to other publications. So, maybe authors can provide additional details about CLoCK study within current manuscript.

Response: Thank you for acknowledging our study as timely and interesting and that we have conducted it with rigour. As requested, we now provide additional details about CLoCK study within the current manuscript so that it can be read as a stand-alone piece although we are limited by space - we now provide much more detail on how (re)infections were excluded and how consent was provided. 

In the Method section, mention the response rate in the recruitment process.

Response: At the end of the 2nd paragraph of “Methods” (lines 150-152) we have now stated: “The final study sample comprised of 3,395 CYP … , representing a 47.8% response rate of participants eligible to be included (100%*3395/7,096).”

In figure 1. The word “reinfected (n-27)” is covered by another box. Please fix it.

Response: We have corrected this formatting error.

Another suggestion would be to have a “Conclusion” section at the end. In the current format, the last paragraph consists of limitations and importance of this study. I would recommend adding the conclusion section and highlighting some of the key findings. The statements in line 351 on page 15 “It is our view that new symptoms…….” And line 355 on page 15 “ However, regardless of whether symptoms…..” are very important statements, so in my opinion they should be part of conclusion section and should be highlighted in the abstract too.

Response: Thank you for this excellent suggestion. We have now edited our manuscript to end with “In conclusion, while acknowledging that (re)infections may have gone undetected, our study demonstrates why new symptoms arising 6-months post-infection should not be exclusively viewed as new long COVID symptoms and should be seen within the wider context of symptomology in the general population. Importantly, regardless of whether symptoms are (or are not) attributable to SARS-COV-2 infection, many CYP experiencing unwanted symptoms that warrant investigation and potential intervention.” (lines 434-439) We have also edited the abstract to read “In CYP, the prevalence of specific symptoms reported at time of PCR-testing declined with time. Similar patterns were observed among test-positives and test-negatives and new symptoms were reported six months post-test for both groups suggesting that symptoms are unlikely to exclusively be a specific consequence of SARS-COV-2 infection. Many CYP experienced unwanted symptoms that warrant investigation and potential intervention.” (lines 78-82)

Reviewer #3 

Thank you for inviting me as a reviewer of this article. I would like to suggest adding following sentence and citation on the "Statistical methods" section. Prevalence N (%) of CYP who had a symptom (never, once, twice or thrice) at testing, at 3 months 211 post-test, at 6 months post-test was tested by Chi-squared test [1] [1] https://doi.org/10.54724/lc.2022.e1

Response: We have added the requested text (with minor edits) and reference to the “Statistical methods”, it now read: “…, we tabulate the prevalence (N (%)) of CYP who had a symptom never, once, twice or thrice (at testing, 3- and 6-months post-test) and assess whether the distribution of symptom prevalence differed by SARS-CoV-2 PCR status using a chi-squared test [18].” (lines 180-182)

---

## [Decision Letter · Decision Letter 1]

2 Nov 2022

Long COVID - Six months of prospective follow-up of changes in symptom profiles of non-hospitalised children and young people after SARS-CoV-2 testing: a national matched cohort study (The CLoCk) Study.

PONE-D-22-21048R1

Dear Dr. Pinto Pereira,

We’re pleased to inform you that your manuscript has been judged scientifically suitable for publication and will be formally accepted for publication once it meets all outstanding technical requirements.

Kind regards,

Dong Keon Yon, MD, FACAAI

Academic Editor

PLOS ONE

Additional Editor Comments (optional):

This is an excellent paper.

Reviewers' comments:

Reviewer's Responses to Questions

**Comments to the Author**

1. If the authors have adequately addressed your comments raised in a previous round of review and you feel that this manuscript is now acceptable for publication, you may indicate that here to bypass the “Comments to the Author” section, enter your conflict of interest statement in the “Confidential to Editor” section, and submit your "Accept" recommendation.

Reviewer #1: (No Response)

Reviewer #3: All comments have been addressed

2. Is the manuscript technically sound, and do the data support the conclusions?

Reviewer #1: Yes

Reviewer #3: Yes

3. Has the statistical analysis been performed appropriately and rigorously? 

Reviewer #1: Yes

Reviewer #3: Yes

4. Have the authors made all data underlying the findings in their manuscript fully available?

Reviewer #1: No

Reviewer #3: Yes

5. Is the manuscript presented in an intelligible fashion and written in standard English?

Reviewer #1: Yes

Reviewer #3: Yes

6. Review Comments to the Author

Reviewer #1: This remains an important study. The authors have responded to the comments that I provided on the original manuscript. They have clarified why they have decided to continue with univariate analyses. This is now a matter of opinion. In my view, the fact that symptomatic clusters might vary across populations is the reason why multivariate analyses are conducted, so that the extent to which a potential syndrome is a generalised phenomenon, with a common origin, may be assessed. However, I do not see this difference in opinion as a reason to hold up publication of a timely study. I just hope that the large data set that has been assembled is analysed in a more nuanced way at some point in the future

Reviewer #3: All comments have been addressed. Thank you to the authors and editors for considering my opinion on this manuscript.

7. PLOS authors have the option to publish the peer review history of their article (what does this mean?). If published, this will include your full peer review and any attached files.

Reviewer #1: No

Reviewer #3: No

---

## [Editor Report · Acceptance letter]

24 Feb 2023

PONE-D-22-21048R1 

Long COVID - Six months of prospective follow-up of changes in symptom profiles of non-hospitalised children and young people after SARS-CoV-2 testing: a national matched cohort study (The CLoCk) Study. 

Dear Dr. Pinto Pereira:

I'm pleased to inform you that your manuscript has been deemed suitable for publication in PLOS ONE. Congratulations! Your manuscript is now with our production department. 

Kind regards, 

on behalf of

Dr. Dong Keon Yon 

Academic Editor

PLOS ONE